# Free Convection and Heat Transfer in Porous Ground Massif during Ground Heat Exchanger Operation

**DOI:** 10.3390/ma15144843

**Published:** 2022-07-12

**Authors:** Borys Basok, Borys Davydenko, Hanna Koshlak, Volodymyr Novikov

**Affiliations:** 1Department of Thermophysical Basics of Energy-Saving Technologies, Institute of Engineering Thermophysics National Academy of Sciences of Ukraine, 03057 Kyiv, Ukraine; basok@ittf.kiev.ua (B.B.); bdavydenko@ukr.net (B.D.); nvg52@i.ua (V.N.); 2Department of Building Physics and Renewable Energy, Kielce University of Technology, 25-314 Kielce, Poland

**Keywords:** porous medium, soil, ground heat exchanger, filtration, heat pump, numerical simulation

## Abstract

Heat pumps are the ideal solution for powering new passive and low-energy buildings, as geothermal resources provide buildings with heat and electricity almost continuously throughout the year. Among geothermal technologies, heat pump systems with vertical well heat exchangers have been recognized as one of the most energy-efficient solutions for space heating and cooling in residential and commercial buildings. A large number of scientific studies have been devoted to the study of heat transfer in and around the ground heat exchanger. The vast majority of them were performed by numerical simulation of heat transfer processes in the soil massif–heat pump system. To analyze the efficiency of a ground heat exchanger, it is fundamentally important to take into account the main factors that can affect heat transfer processes in the soil and the external environment of vertical ground heat exchangers. In this work, numerical simulation methods were used to describe a mathematical model of heat transfer processes in a porous soil massif and a U-shaped vertical heat exchanger. The purpose of these studies is to determine the influence of the filtration properties of the soil as a porous medium on the performance characteristics of soil heat exchangers. To study these problems, numerical modeling of hydrodynamic processes and heat transfer in a soil massif was performed under the condition that the pores were filled only with liquid. The influence of the filtration properties of the soil as a porous medium on the characteristics of the operation of a soil heat exchanger was studied. The dependence of the energy characteristics of the operation of a soil heat exchanger and a heat pump on a medium with which the pores are filled, as well as on the porosity of the soil and the size of its particles, was determined.

## 1. Introduction

Today, the active use of renewable energy sources in the energy supply systems of buildings is the main component of energy-efficient and passive construction projects. Among geothermal technologies, the use of a heat pump system with vertical borehole heat exchangers has become the focus of many researchers. Ground source heat pump systems use the stable ground temperature as a source or sink of heat, which is higher and lower than the air temperature in winter and summer, respectively, and therefore, such technologies can provide better thermal efficiency than normal air source heat pumps [1]. However, there are some unsolved problems in ground source heat pump technology. Several reasons for this can be named. First, the problems of coordinating the implementation of the thermodynamic cycle of a heat pump under conditions of constant load on the ground heat exchanger and a variable load on the heat supply system have not been fully solved. Secondly, when designing a geothermal heat pump and determining the dimensions of a ground heat exchanger, one of the problems is the difficulty in determining in detail the thermal characteristics of the ground medium, which are influenced by several factors and, above all, climatic conditions. The authors of [2] concluded that the simplifying assumptions for analytical calculations of the soil temperature distribution are unrealistic, since the change in the properties of the earth’s surface by months of the year has a significant impact on the temperature distribution. Bloom et al. reviewed the technical and economic factors affecting the design and performance of vertical geothermal heat pump systems, and assessed the spatial correlation of these factors with geographic components such as geology and climatic conditions. According to their research so far, subsurface characteristics are not adequately considered during the planning and design of small-scale GSHP systems, which causes under- or over sizing and, therefore, a long-term impact on the maintenance costs and payback time of such systems [3]. Several factors, such as the amount of precipitation, air temperature, windiness, and insolation, determine the potential ability of a low-potential heat source to accumulate energy. Other factors affecting the intensity of absorption and accumulation of energy are the composition and properties of the upper soil layer, as well as its structure and moisture. It was noted in [4,5] that the average monthly air temperature and seasonality affect the temperature of soil layers at a depth of more than 1 m, but the soil temperature remains almost constant at a depth of more than 10 m [6]. However, vertical heat exchangers have a much greater installation depth, at which changes in the heat consumption of the soil are also possible, for example, in the presence of convective groundwater flows. In this case, one cannot simply enter the heat amount values recommended for calculations that can be obtained from the soil, since the convective flow significantly increases this potential (for example, up to 140 W/m^2^ with an average recommended value of 50 W/m^2^). Thus, in conditions of absolute instability of the parameters that the thermodynamic cycle (Hampson–Linde cycle) must provide, the change in the thermodynamic parameters of the heat pump cycle cannot always be compensated by the controls provided for the heat pump. When setting the efficiency parameters of the heat pump system, for example, COP = 5.6, changing the value of low-potential energy can significantly reduce this indicator.

Theoretically, it is believed that an increase in the value of low-potential energy input contributes to an increase in COP. In practice, when reducing the heat demand in a house, one has to use various approaches to reduce the temperature of the superheated thermodynamic carrier, for example, by injecting cold vapor behind the condenser, into the compressor (if the heat pump is equipped with this technology), which reduces COP by up to 30%, i.e., to 3.9 instead of 6. Therefore, when designing heat pump systems and vertical ground heat exchangers, it is important to take into account the thermal properties of the soil and the characteristics of heat transfer in the ground, for example, convective flux in the presence of groundwater, and to perform design calculations of the ground heat exchanger in more detail.

## 2. Analysis of Research Results and Publications

It is a well-known fact that the U-tube vertical ground heat exchanger is a simple and reliable design [7]. The problem of heat transfer intensification between vertical ground heat exchangers and soil has been studied by numerous researchers [8,9,10] and, first of all, by numerical modeling [11,12,13] of heat transfer processes in the soil mass during the operation of a heat pump. When modeling in calculation modules, heat transfer is usually divided into heat transfer inside the ground heat exchanger and heat transfer outside it. To analyze the efficiency of the ground heat exchanger, it is important to take into account the main factors that can affect the heat transfer processes in the heat exchanger and in the environment (soil). These factors include the composition of the cement mortar material [14], layout of the tubes [15], recovery time, depth and diameter of the well [16,17], and soil properties [18]. In case of water flow in the soil, the heat transfer from the circulating fluid to the ground heat exchanger tube can be considered predominantly convective. In the area from the tube of the heat exchanger to the soil, conductive heat transfer prevails.

In [19], the results of computer simulation of the dynamics of the processes of accumulation and extraction of heat in a soil mass with a single vertical heat exchanger are presented. The calculation results are compared with analytical and computational models. Satisfactory agreement between the results is obtained. Study [20] numerically investigates the process of heat transfer in a soil mass containing a horizontal ground heat collector. Temperature conditions for operation are determined for a working ground heat exchanger on the basis of calculations. The model of heat transfer in vertical ground heat exchangers for heat pump systems is also considered in [21]. The solution to the problem of heat transfer in the soil mass is obtained in an explicit form. The resulting mathematical formula satisfactorily describes the temperature regime of the heat exchanger and can be included in software used for the thermal analysis of ground heat exchangers. It also presents a model of heat transfer inside the borehole, taking into account the thermal interaction between the supports of the U-tube. This can be used to build a formula for thermal resistance inside the borehole.

In [22], a three-dimensional model of heat transfer in these systems is presented, using the finite volume method for implementation purposes. The proposed model takes into account the interconnected processes of heat transfer in the tubes and the soil, which is located between the tubes. Comparison of the calculation results for this model with experimental data shows that the model provides a sufficiently high accuracy.

In [23], the results of computer simulation of borehole ground heat exchangers used in geothermal heat pump systems are presented. The results are based on a 3D model using the implicit finite difference method in a rectangular coordinate system. Each well is approximated by a square column bound by the radius of the borehole. To solve the system of difference equations, the iteration method is used at each step. A comparison is made of the results of calculations by this method and by the model of a finite linear source. The discrepancies between the results obtained by the two methods increase together with the size of the borehole.

In [24], a three-dimensional computational hydrodynamic model of a ground source heat pump with several energy accumulators is presented. The model is designed to investigate the heating performance of a system under continuous and intermittent operating conditions and evaluate the system’s thermal energy recovery and performance indicators. The 3D model is based on hybrid meshes with unstructured and structured types of tetrahedra and hexagons. Satisfactory agreement between the results of CFD modeling and experimental data is achieved. The study demonstrates that the soil temperature in the intermittent operation mode is higher than in the continuous operation mode of the heat pump. It has been established that intermittent operation not only helps to restore soil temperature, but also improves the overall performance of the system.

In the works cited, the soil is considered to be a continuous medium. It is believed that heat transfer occurs only through the thermal conductivity of the soil and depends on the thermal resistance of the heat exchanger [25,26]. In reality, the soil is a porous medium, and its pores can be filled with air and fluids [27,28,29]. In this regard, in addition to thermal conductivity, heat transfer in the soil can also occur by convection of fluids or gases in a porous medium. Convection can be either natural (due to the presence of a temperature gradient in the mass) or forced (or mixed) in case of a pressure gradient in the soil mass. Models of mass, momentum, and heat transfer through a porous medium are being developed to numerically study this problem. In [30], to assess the effect of groundwater flow on the operation of geothermal heat exchangers in heat pump systems with a ground heat source, the heat transfer equation is applied, taking into account advection in a porous medium, and its analytical solution is obtained based on Green’s function method. This method is used to determine the effect of groundwater advection on heat transfer. Calculations show that water advection in a porous medium can significantly change the temperature distribution in the soil mass. The hydraulic and thermal properties of soils and rocks affecting the transfer of heat by advection are described.

In [31], the influence of natural convection on the operation of heat exchangers in closed-loop geothermal systems is studied. For numerical study of this problem the Darcy–Brinkman–Forchheimer model [32,33] is used. The basic equations of continuity, momentum, and energy balance are derived taking into account the porosity of a soil medium completely saturated with fluid. The flow of fluid in the pores occurs due to natural convection. The discretization method on a structured grid is used for solving the basic equations. The performance of the heat exchanger is estimated by the volume of extracted energy and by the temperature of the heat carrier at the outlet of the heat exchanger. The results are evaluated by comparing them with the results of calculations according to known existing heat transfer models applied for heat transfer only by thermal conduction. The effect of natural convection and the filtration characteristics of the soil on the operation parameters of the heat exchange unit is determined.

Models of flow in porous media are used not only in problems of free convection in a soil massif. They are also used in modeling the flow of nanofluids [34], including micropolar ferrofluids [35].

The analysis of studies devoted to the issue of heat transfer in a soil porous medium demonstrates that most of the applied models of ground heat exchangers operation assume that the soil is a continuous medium with known thermophysical properties and that heat transfer in the soil occurs only through heat conduction. Heat transfer models that consider heat transfer only by thermal conductivity are adequate in cases where convection affects the total volume of heat extracted from the soil to a much lesser extent. This takes place when the permeability of the porous medium for gas or fluid is sufficiently small. If the soil permeability is significant, it is necessary to apply heat transfer models that take into account the presence of filtration transfer. These issues should be investigated in more detail to determine the characteristics of fluid flow and heat transfer in a porous soil mass during the operation of ground heat exchangers.

## 3. Purpose of the Study

The purpose of this study is to determine the influence of the filtration properties of the soil as a porous medium on the performance characteristics of ground heat exchangers. For the computational study of these issues, numerical simulation of hydrodynamics and heat transfer in the soil mass was performed, with the assumption that the pores are filled only with fluid; that is, the case of single-phase filtration of a substance in a soil mass during the operation of a heat pump installation is considered.

## 4. Statement of the Problem of Heat Transfer in a Soil Mass in the Presence of Filtration Processes

The problem of the temperature state of the soil mass during the operation of the ground heat exchanger is formulated as follows. The process of heat transfer in the computational domain, which has the shape of a rectangular parallelepiped with sides *x_max_*, *y_max_*, and *z_max_*, is considered. This parallelepiped covers a section of the soil mass with a vertical U-tube heat exchanger filled with a fluid heat carrier circulating through it. The scheme of the computational domain is shown in Figure 1.

The values *x_max_*, *y_max_*, and *z_max_* are chosen so that the processes of heat transfer to the ground heat exchanger have a minimal effect on the temperature conditions at the boundaries of the computational domain. Soil is considered to be a porous medium, assuming the pores are filled with water.

As a result of the difference between the temperature of the heat carrier in the heat exchanger and the temperature of the soil, a free-convection flow of the fluid occurs in the soil mass, between the solid particles of the soil. For simulation of the flow of a fluid in a porous medium, the Darcy–Brinkman–Forchheimer model [32] is used. This flow is described by a system of equations, which includes:–Continuity equation:
(1)$$\frac{\partial u}{\partial x}+\frac{\partial \upsilon }{\partial y}+\frac{\partial w}{\partial z}=0;$$–Momentum equations:
(2)$$\rho_{f}\left(\frac{1}{\varphi }\frac{\partial u}{\partial \tau }+\frac{u}{\varphi^{2}}\frac{\partial u}{\partial x}+\frac{\upsilon }{\varphi^{2}}\frac{\partial u}{\partial y}+\frac{w}{\varphi^{2}}\frac{\partial u}{\partial z}\right)=-\frac{\partial p}{\partial x}+\frac{\mu }{\varphi }\nabla^{2}u-\frac{\mu }{K}u-\frac{\rho_{f}c_{F}}{\sqrt{K}}\left|V\right|u$$
(3)$$\rho_{f}\left(\frac{1}{\varphi }\frac{\partial \upsilon }{\partial \tau }+\frac{u}{\varphi^{2}}\frac{\partial \upsilon }{\partial x}+\frac{\upsilon }{\varphi^{2}}\frac{\partial \upsilon }{\partial y}+\frac{w}{\varphi^{2}}\frac{\partial \upsilon }{\partial z}\right)=-\frac{\partial p}{\partial y}+\frac{\mu }{\varphi }\nabla^{2}\upsilon -\frac{\mu }{K}\upsilon -\frac{\rho_{f}c_{F}}{\sqrt{K}}\left|V\right|\upsilon$$
(4)$$\rho_{f}\left(\frac{1}{\varphi }\frac{\partial w}{\partial \tau }+\frac{u}{\varphi^{2}}\frac{\partial w}{\partial x}+\frac{\upsilon }{\varphi^{2}}\frac{\partial w}{\partial y}+\frac{w}{\varphi^{2}}\frac{\partial w}{\partial z}\right)=-\frac{\partial p}{\partial z}+\frac{\mu }{\varphi }\nabla^{2}w-\frac{\mu }{K}w-\frac{\rho_{f}c_{F}}{\sqrt{K}}\left|V\right|w-g\beta \left(t_{p}-t_{\infty }\right)$$–Energy equation:
(5)$$C_{p}\rho_{p}\left(\frac{\partial t_{p}}{\partial \tau }+u\frac{\partial t_{p}}{\partial x}+\upsilon \frac{\partial t_{p}}{\partial y}+w\frac{\partial t_{p}}{\partial z}\right)=\lambda_{p}\nabla^{2}t_{p}$$
where $\nabla^{2}=\frac{\partial^{2}}{\partial x^{2}}+\frac{\partial^{2}}{\partial y^{2}}+\frac{\partial^{2}}{\partial z^{2}}$—Laplace operator; |*V*|—fluid flow velocity vector modulus. The vertical coordinate *z* is directed from the soil surface (*z* = 0) down. Thermophysical properties of the porous medium are calculated using formulas:$$\lambda_{p}=\varphi \lambda_{f}+\left(1-\varphi \right)\lambda_{s};$$
$$C_{p}\rho_{p}=\varphi C_{f}\rho_{f}+\left(1-\varphi \right)C_{s}\rho_{s}.$$

To determine the coefficients *K* and *c_F_*, the following relationships (6) and (7) are accepted, indicating their dependence on the porosity of the material *φ* and the diameter of the soil particles *d_p_*
(6)$$K=\frac{{d_{p}}^{2}}{150}\frac{\varphi^{3}}{{\left(1-\varphi \right)}^{2}};$$
(7)$$c_{F}=\frac{1.75}{\varphi^{3/2}150^{1/2}}$$

The boundary conditions for the system of Equations (1)–(5) are the following:$$x=0;\;x=x_{\max}:u=0;\;\frac{\partial \upsilon }{\partial x}=0;\;\frac{\partial w}{\partial x}=0;t=t_{\infty };$$
$$y=0;\;y=y_{\max}:\upsilon =0;\;\frac{\partial u}{\partial y}=0;\;\frac{\partial w}{\partial y}=0;t=t_{\infty };$$
$$z=0:w=0;\;\frac{\partial u}{\partial z}=0;\;\frac{\partial \upsilon }{\partial z}=0;\;\frac{\partial t}{\partial z}=0;$$
$$z=y_{\max}:w=0;\;\frac{\partial u}{\partial z}=0;\;\frac{\partial \upsilon }{\partial z}=0;t=t_{\infty }.$$

As follows from the given boundary conditions for *z* = 0, heat transfer from the outer surface of the soil is not taken into account. To simplify the problem, the heat exchanger channel is represented by two straight vertical sections connected by a horizontal section. Sections of the channel are considered to be square. The length of the sides of the square is *a*, and the thickness of the channel walls is δ.

For the heat carrier flow in the heat exchanger channel, the energy equation has the following form:–For the section of the downward flow in the vertical section of the channel:
(8)$$C_{c}\rho_{c}\left(\frac{\partial t_{c}}{\partial \tau }+U\frac{\partial t_{c}}{\partial z}\right)=\lambda_{c}\nabla^{2}t_{c};$$–For the upstream section in the vertical section of the channel:
(9)$$C_{c}\rho_{c}\left(\frac{\partial t_{c}}{\partial \tau }-U\frac{\partial t_{c}}{\partial z}\right)=\lambda_{c}\nabla^{2}t_{c};$$–For the horizontal section of the channel:
(10)$$C_{c}\rho_{c}\left(\frac{\partial t_{c}}{\partial \tau }+U\frac{\partial t_{c}}{\partial x}\right)=\lambda_{c}\nabla^{2}t_{c};$$
where $U=\frac{G}{a^{2}}.$

On the outer surface of the heat exchanger channel in contact with the soil, the following conditions are assumed:(11)$$u=0;\;\upsilon =0;\;w=0;-\lambda_{p}\frac{\partial t_{p}}{\partial n}=\frac{t_{p}-t_{c}}{\frac{\delta }{\lambda_{w}}+\frac{1}{\alpha_{c}}},$$
where *n* is the direction of the outer normal to the outer surface of the channel; α*_c_* is the heat transfer coefficient in the heat exchanger channel, determined by the formula given in [31]: $\alpha_{c}=3.66\cdot \lambda_{c}/D_{e}$; $D_{e}$ is the equivalent diameter of a square channel.

The system of Equations (1)–(10) with the given boundary conditions is solved using the finite difference method. To solve the system of difference equations of fluid dynamics in a porous medium (1)–(4), the SIMPLE algorithm [32] is used. To solve the energy Equation (5) for a porous medium, together with Equations (8)–(10) for the heat carrier and the conjugation condition (11), an explicit time scheme is used.

## 5. Results of Numerical Studies and Their Analysis

As an example, a ground heat exchanger is considered, the cross-section of which is a square with the side *a* = 0.1 m. The thickness of the channel walls is 2 mm. Its total length is *L* = 32.67 m. The material of the heat exchanger channel is polyethylene. Heat carrier consumption is *G* = 0.21 × 10^−3^ m^3^/s. The heat carrier is an aqueous solution of polypropyleneglycol. Its flow velocity is *U* = 0.021 m/s. The time period during which the heat carrier flows from the inlet of the channel to the outlet is Δτ = *L*/*U* = 1556 s (25.9 min). At the initial moment, the temperature of the soil mass is *t*_∞_ = 10 °C. During 45 min, the temperature of the heat carrier at the inlet to the heat exchanger is *t*_c_ = 5 °C. Further, during the next 15 min, the cooling of the heat carrier in the heat pump stops, and the heat carrier enters the heat exchanger with a temperature of *t*_c_ = 10 °C. After 15 min, the heat carrier cooling cycle is repeated. Two variants of soil porosity are considered: *φ* = 0.48 and *φ* = 0.40 with soil particle diameter of *d_p_* = 0.5 mm. The pores contain water. Thermal conductivity for solid particles is taken *λ*_s_ = 1.5 W/m/K and for water—*λ_f_* = 0.58 W/m/K [31]. At the boundaries of the computational domain, the temperature is maintained at the level *t*_∞_ = 10 °C.

The results of the calculation of the temperature regime of the ground heat exchanger are presented in the form of fluid velocity fields in the pores and temperature fields. Velocity and temperature fields in the vertical section of the porous soil mass for the case *φ* = 0.48; *d_p_* = 0.5 mm are presented in Figure 2. The free-convection flow of ground water in the pores arises due to the difference between the temperatures of the heat carrier and the soil mass. As can be seen from the figure, the water flow in the pores near the cooled surface of the heat exchanger channel is directed downward. On the borders of the calculation area, the flow of water in the pores is directed upwards.

To determine the effect of the medium filling the soil pores on the efficiency of the ground heat exchanger, the problem solved for the case of pores filled with water is also solved for the case of pores filled with air. Thermal conductivity for air is *λ_f_* = 0.026 W/m/K.

The distribution of the vertical velocity *w_z_* horizontally in the direction of the 0X axis along the line intersecting the heat exchanger at a depth of 9.0 m is presented in Figure 3a. The temperature distribution in the soil mass along this line is presented in Figure 3b. Curves 1 refer to the case when the pores are filled with water, and Curves 2 refer to the case when the pores are filled with air. From Figure 3a (Curve 1), it can be seen that the maximum value of 1.6 × 10^−6^ m/s for velocity *w_z_* is observed near the surfaces of the channel. For the case of filling the pores with air, the maximum velocity at the outer and inner surfaces of the channel is much higher and amounts to *w_z_* ~4.7 × 10^−6^ m/s (Curve 2).

Changes in time of the heat carrier temperature at the inlet to the heat exchanger, which is specified, and the heat carrier temperature at the outlet of the heat exchanger, which is determined from the calculation, are presented in Figure 4. Data refer to the case of filling the pores with water at *φ* = 0.48; *d_p_* = 0.5 mm.

Curve 1 shows the temperature of the heat carrier at the inlet to the heat exchanger, and Curve 2 shows the temperature of the heat carrier at the outlet of the heat exchanger. As can be seen from these figures, at the beginning of each hour, the temperature of the heat carrier at the inlet to the heat exchanger channel is 5 °C. The temperature at the outlet of the channel rises relative to the temperature at the inlet. Last 15 min. of every hour the temperature of the heat carrier at the inlet to the heat exchanger rises to *t*_c_ = 10 °C, i.e., the heat carrier during these 15 min. not cooled in the heat pump. In this case, the time period during which the heat carrier that entered the heat exchanger is completely removed from it is Δτ = 25.9 min. The period of time during which the temperature of the heat carrier at the inlet to the heat exchanger rises to a temperature of *t*_c_ = 10 °C is only 15 min. Therefore, the heat exchanger channel simultaneously contains both cooled and uncooled heat carriers. As a result, the maximum values of the heat carrier temperature at the outlet of the heat exchanger lag significantly in time from the maximum temperature values of the heat carrier at the inlet to the heat exchanger.

To determine the effect of soil porosity on the energy characteristics of a ground heat exchanger, similar calculations are also performed for the case *φ* = 0.40; *d_p_* = 0.5 mm. Figure 5 shows the difference between the temperature values at the outlet of the ground heat exchanger for the case of *φ* = 0.40 and *φ* = 0.48 when the pores are filled with water. This figure shows that the temperature of the heat carrier at the outlet of the heat exchanger channel for the case *φ* = 0.40 is 0.002…0.016 °C higher than for the case *φ* = 0.48; that is, a greater volume of extracted heat is provided with less porosity.

The heat carrier temperature distributions along the heat exchanger channel for the cases *φ* = 0.48 and *φ* = 0.40 at filling the pores with water and air are presented in Figure 6. As can be seen from this figure, for the case of *φ* = 0.40, the temperature at the exit from the heat exchanger is somewhat higher than for the case of *φ* = 0.48. From this, it follows that a larger volume of extracted heat is provided at a lower porosity. For the case of filling the pores with water, the volumes of heat withdrawn from the soil mass are 219.81 W at *φ* = 0.48 and 231.27 W at *φ* = 0.40. The difference between these heat volumes is 11.46 W.

Due to the fact that the velocity of free-convection flow of the fluid in the pores under these conditions is very small, the flow of the fluid does not significantly affect the heat transfer from the ground to the heat exchanger. Therefore, the effect of porosity on the intensity of heat transfer is manifested due to the change in the thermophysical properties of the soil with a change in porosity.

## 6. Discussion

As follows from the comparison of the data presented in Figure 3a, referring to the cases of pores filled with water (Curve 1) and with air (Curve 2), the nature of the velocity distribution for the case when the pores are filled with air is qualitatively similar to the case when the pores are filled with water. However, in quantitative terms, the maximum flow velocity when the pores are filled with air is three times higher than in the case of pores filled with water.

As can be seen from the comparison of Curves 1 and 2 in Figure 6, relating to the case *φ* = 0.48, for the case of pores filled with water, the temperature at the outlet from the heat exchanger channel is approximately 0.075 °C higher than for the case of pores filled with air. It follows from this that a greater volume of extracted heat is provided when the pores are filled with water. For the conditions under consideration, the volumes of heat extracted from the soil mass are 219.81 W when the pores are filled with water (*d_p_* = 0.5 mm; *φ* = 0.48) and 165.27 W when the pores are filled with air. The difference between these heat volumes is 54.54 W. For the case *d_p_* = 0.5 mm; *φ* = 0.40, the difference between the volumes of heat extracted from the soil when the pores are filled with water and air is 48.0 W.

To determine the impact of the medium filling the soil pores on the energy characteristics of the heat exchanger, the difference between the temperature values of the heat carrier at the outlet of the soil heat exchanger for the cases of pores filled with water and filling with air (*d_p_* = 0.5 mm; *φ* = 0.48) is calculated. The change in time for this difference is presented in Figure 7. Figure 7 shows that the temperature of the heat carrier at the outlet of the heat exchanger in the case of pores filled with water is 0.01…0.08 °C higher than in the case of pores filled with air. This also confirms the fact that the energy characteristics of the heat exchanger are higher when the pores are filled with water compared to when the pores are filled with air.

The novelty of the presented results is that they, in contrast to most published works on the problem of natural convection in soil, were obtained on the basis of solving a three-dimensional problem of free convection for two types of media (water and air) filling soil pores. A porous medium (soil) was represented as solid spherical particles of a given diameter. Two variants of soil porosity were considered. The problem was solved for the area where the vertical ground heat exchanger is located. In fact, the studies were carried out for wet and absolutely dry soil. The physical model is based on the main manifestations of the filtration process—the Darcy, Brinkman, and Forchheimer effects. These calculations were performed for realistic heat pump operation modes—three quarters of the cycle the compressor is working, one quarter is not. The main result of numerical simulation is that, with the same geometry of the heat exchanger, its energy efficiency is higher with less soil porosity. In addition, the heat exchanger works more efficiently in wet soil. This is due to the influence of the thermal conductivity coefficient, the value of which increases with a decrease in soil porosity and in the presence of moisture. But even in absolutely dry soil, it functions quite rationally. Its thermal efficiency is reduced by only 27% compared to wet ground. Due to the low velocity (~10^−6^ m/s) of the free-convective flow of media filling the pores, convection under these conditions has little effect on the heat transfer.

The complexity of the practical engineering application of this model lies in the fact that in typical soils with low humidity, the thermal performance of such a single U-tube heat exchanger is not high enough at a depth range of up to 100 m. Therefore, it is advisable to organize a group of such devices to achieve the required thermal power of 0.5–1 MW due to the natural heat of the soil. This raises the problem of optimizing the mutual arrangement of heat exchangers in the group. It is necessary to determine the distances between single heat exchangers, the geometry of their arrangement, and the depth of well drilling. In the problem of the accumulation of heat in the soil and its extraction from the soil, other time intervals arise—seasonal, monthly (ten-day), or daily. These intervals determine the time for the accumulation of heat and for its subsequent extraction.

The proposed numerical model and the results of calculations for this model can be used in the design:–Systems for low-temperature ground heat extraction for “green” heat pumps;–Systems of forced accumulation and extraction at the required moment of time of the injected heat, for example, in the conditions of off-season storage of heat or cold;–Systems that do not use a compressor for air conditioning (for example, a cold-water floor);–Geothermal ventilation systems;–Heat and power systems of passive buildings, “zero-energy” buildings;–Air thermal curtain systems for ventilated building facades.

Some of these problems are already being solved [19,20,33,34,35].

## 7. Conclusions

Based on the results of the numerical solution of the system of equations concerning the fluid dynamics and heat transfer in a porous medium filled with water and air, the characteristics of the free-convection fluid flow in a soil mass in the presence of a vertical U-tube ground heat exchanger were obtained. Free-convection flow occurs when there is a temperature gradient in the soil mass, which is a consequence of the operation of the heat pump, one component of which is the ground heat exchanger. The heat pump operates in an intermittent mode, which results in a change in the time of the temperature state of the soil mass and the energy performance of the heat exchanger.

The distributions of temperature and velocity of free-convection flow in a porous soil mass were obtained using computational methods. The results obtained for different values of soil porosity were compared. It was demonstrated that the maximum velocity of free-convection water flow in pores under the considered conditions is of the order of ~10^−6^ m/s. Under these conditions, the effect of natural convection on the heat transfer in the soil mass Is insignificant.

An assessment of the energy performance of the ground heat exchanger, depending on the porosity of the soil, showed that with the remaining operating parameters of the ground heat exchanger being equal, a greater volume of heat extracted from the soil is provided in cases of lower porosity. It also follows from the analysis of the energy characteristics of the heat exchanger operation that these characteristics are higher when the pores are filled with water than when the pores are filled with air; that is, when the pores are filled with water, a greater volume of heat is extracted from the soil compared to the case when the pores are filled with air.

In the future, this numerical model of heat transfer in soil under forced convection of a liquid in a porous medium is supposed to be used in the design of a soil heat exchanger located completely or partially in an aquifer. It is likely that the presence of an aquifer will significantly increase the thermal performance of the heat exchanger. Also of interest is the calculation of heat transfer in the “soil–multipass heat exchanger” system, in which atmospheric air is used as a heat carrier. Such systems are used for geothermal ventilation of the building. Geothermal ventilation is a modern innovative trend in building construction.

## Figures and Tables

**Figure 1 materials-15-04843-f001:**
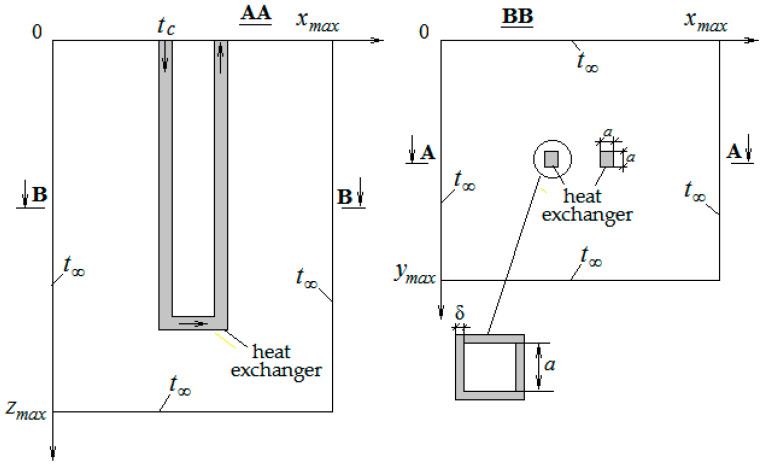
Scheme of the computational domain.

**Figure 2 materials-15-04843-f002:**
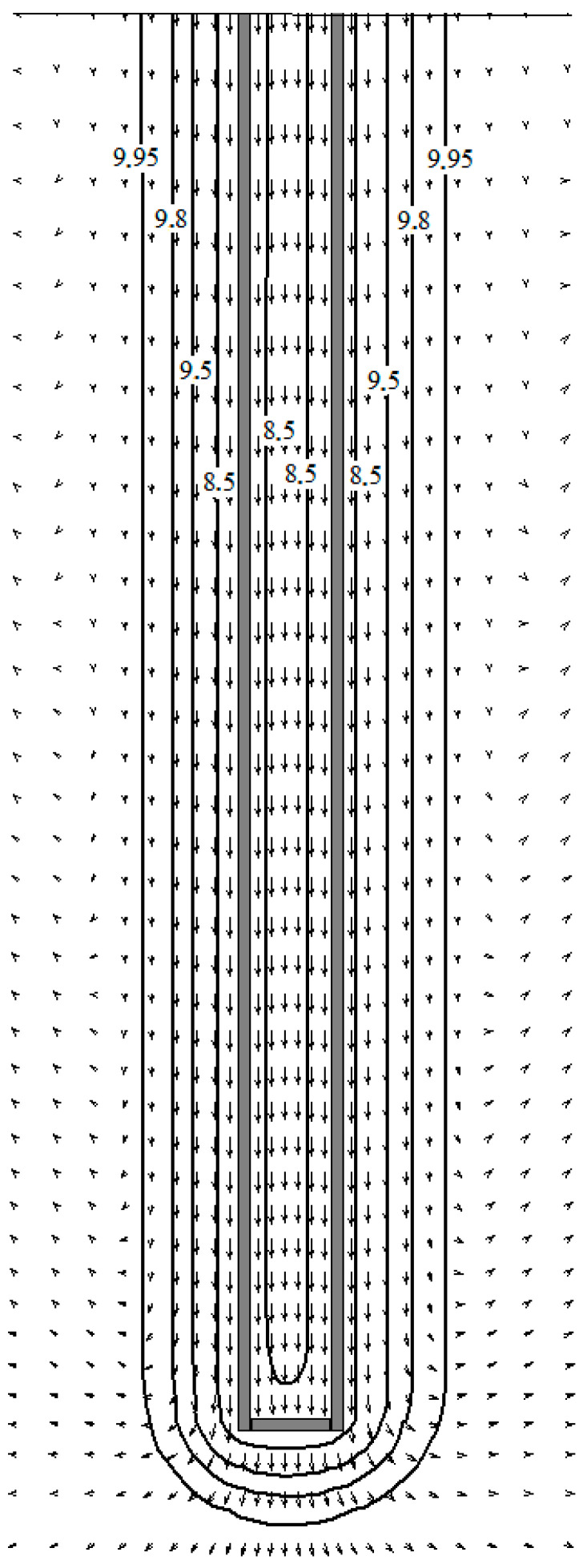
Velocity and temperature distribution (°C) in the soil mass when the pores are filled with water (*d_p_* = 0.5 mm, *φ* = 0.48). The direction of the vectors coincides with the direction of movement, and the length of the vector is proportional to the fluid velocity.

**Figure 3 materials-15-04843-f003:**
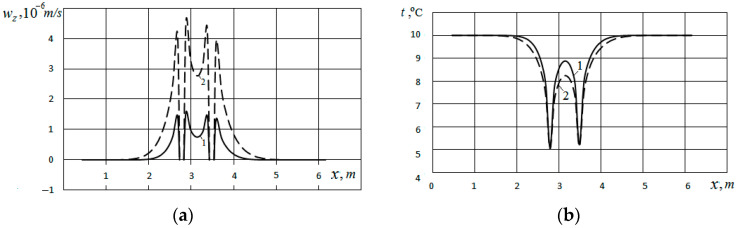
Distribution of vertical velocity (**a**) and temperature (**b**) horizontally at a depth of 9.0 m for the cases of filling pores with water (Curves 1) and with air (Curves 2) at *d_p_* = 0.5 mm; *φ* = 0.48.

**Figure 4 materials-15-04843-f004:**
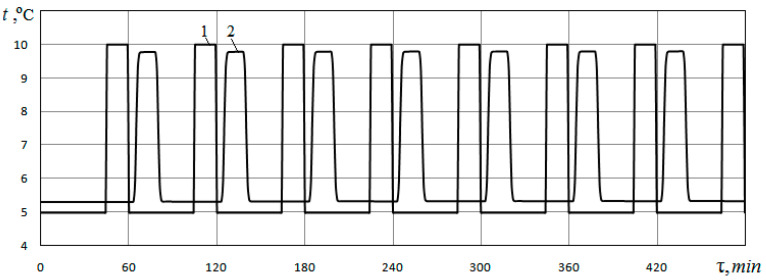
Change in time of the temperature of the heat carrier at the inlet (1) and outlet (2) of the ground heat exchanger for the case of soil porosity *φ* = 0.48 (*d_p_* = 0.5 mm). The pores are filled with water.

**Figure 5 materials-15-04843-f005:**
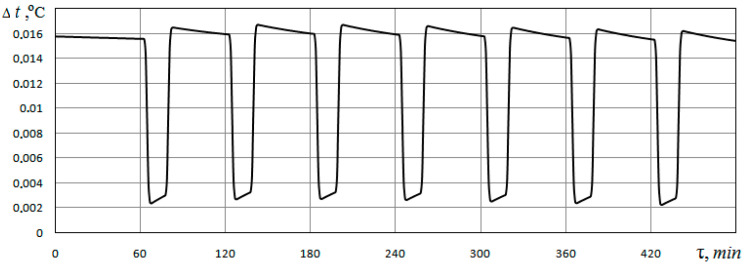
The difference between the temperature values at the outlet of the ground heat exchanger for cases of soil porosity *φ* = 0.40 and *φ* = 0.48, assuming the pores are filled with water.

**Figure 6 materials-15-04843-f006:**
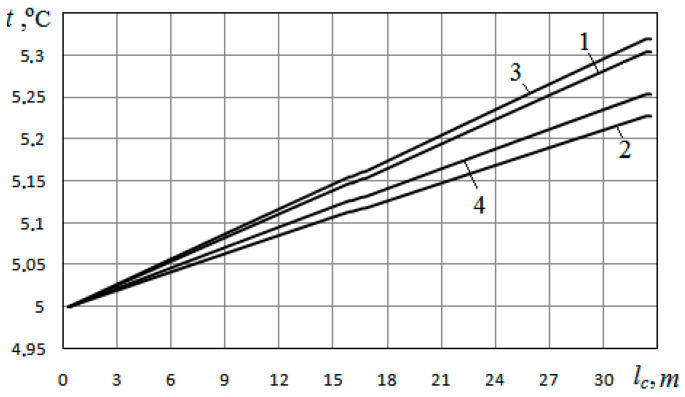
Change in the temperature of the heat carrier along the length of the heat exchanger channel at *d_p_* = 0.5 mm for *φ* = 0.48 (1, 2) and *φ* = 0.40 (3, 4): 1; 3—pores filled with water; 2; 4—pores filled with air.

**Figure 7 materials-15-04843-f007:**
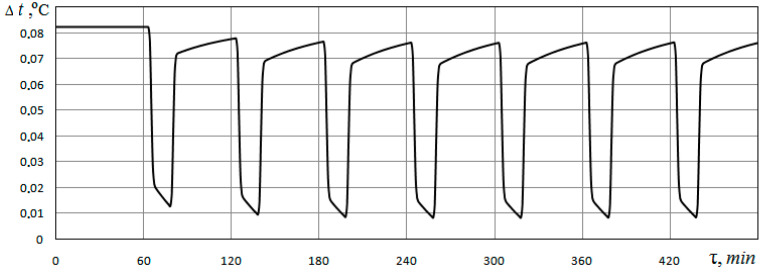
Difference between the values of the heat carrier temperature at the outlet of the ground heat exchanger for the cases of pores filled with water and pores filled with air (*φ* = 0.48; *d_p_* = 0.5 mm).

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
