# Peer review of "Free Convection and Heat Transfer in Porous Ground Massif during Ground Heat Exchanger Operation"

_materials, 2022, doi:10.3390/ma15144843_

Round 1
Reviewer 1 Report
The purpose of this study is to determine the influence of the filtration properties of 165 the soil as a porous medium on the performance characteristics of ground heat exchang-166 ers. For the computational study of these issues, numerical simulation of hydrodynamics and heat transfer in the soil mass was performed, with the assumption that the pores are filled only with fluid. That is, the case of single-phase filtration of a substance in a soil 169 mass during the operation of a heat pump installation is considered. The Darcy-Brink-170 man-Forchheimer model is used to describe the fluid flow in a porous medium. 171
Comments:
Abstract
The results of the research are very general.
Line 179 - 181
The values xmax; ymax and zmax are chosen so that the processes of heat transfer to the ground heat exchanger have a minimal effect on the temperature conditions at the boundaries of the computational domain.
Literature source?
As a result of the difference between the temperature of the heat carrier in the heat exchanger and the temperature of the soil, a free convection flow of the fluid occurs in the soil mass, between the solid particles of the soil. - vague
the computational domain – a sketch should be given
As follows from the given boundary conditions for z=0, heat transfer from the outer surface of the soil is not taken into account. To simplify the problem, the heat exchanger channel is represented by two straight vertical sections connected by a horizontal section. Sections of the channel are considered to be square. The length of the sides of the square is a, the thickness of the channel walls is δ.
- It would be good to give a sketch.
Line 252 - 254
To solve the system of difference equations of fluid dynamics in a porous medium (1)…(4), the SIMPLE algorithm [56] is used. To solve the energy equation (5) for a porous medium, together with equations (8)...(10) for the heat carrier and the conjugation condition (11), an explicit time scheme is used.
- No further information on the software used, etc.
The description of Figure 2 is very brief
Line 338 – 376
- A comparison is given when the pores are filled with air and water. There is talk of convective heat transfer. However, nowhere is the influence of the thermal conductivity of air and water on the exchanged heat explained.
- Darcy, Reynolds, Nuselt numbers are not listed
Author Response
We appreciate the time and effort the reviewer has devoted to providing valuable feedback on our scientific results. Detailed comments added in pdf file below.

Reviewer 2 Report
In this work, numerical simulation methods were used to describe a mathematical model of heat transfer processes in a porous soil massif and a U-shaped vertical heat exchanger. The purpose of these studies is to determine the influence of the filtration properties of the soil as a porous medium on the performance characteristics of soil heat exchangers. To study these problems, numerical modeling of hydrodynamic processes and heat transfer in the 26 soil massif was performed under the condition that the pores are filled only with liquid
1. Summarize the main contribution of each referenced paper in a separate sentence and by including the reference number.
2. The originality of the paper needs to be stated clearly. It is of importance to have sufficient results to justify the novelty of a high-quality journal paper.
3. Include details for equations 1-11, define all terms.
4. Governing equations should be including relevant references.
5. Fluid in a porous medium with the Darcy-Brinkman-Forchheimer model should need a help of references….https://doi.org/10.1016/j.csite.2021.101728,https://doi.org/10.1016/j.icheatmasstransfer.2019.104385
6. Line208,CF will be Cf, line 249 α c=3.66.
7. Numerical section and producer is missing
8. The authors should focused on the discussion section, many sentence are meaningless. Discussion section should be improved, It should not be just increase/decrease include some real discussion, the discussion should include examples of some real applications.Discuss all parameter with specific reasons
9. Some symbols are misplaces in some equations, carefully revise and correct
10. Proper validation of the model is needed also supported by past studies.
Author Response
We appreciate the time and effort that the reviewer has dedicated to providing his valuable feedback on our scientific results. Detailed comments are presented in the PDF file below.

Round 2
Reviewer 1 Report
The authors have made the requested corrections.
Reviewer 2 Report
I recommend to be published